 **eLIFE**

POINT OF VIEW

# Priority of discovery in the life sciences

**Abstract** The job of a scientist is to make a discovery and then communicate this new knowledge to others. For a scientist to be successful, he or she needs to be able to claim credit or priority for discoveries throughout their career. However, despite being fundamental to the reward system of science, the principles for establishing the "priority of discovery" are rarely discussed. Here we break down priority into two steps: disclosure, in which the discovery is released to the world-wide community; and validation, in which other scientists assess the accuracy, quality and importance of the work. Currently, in biology, disclosure and an initial validation are combined in a journal publication. Here, we discuss the advantages of separating these steps into disclosure via a preprint, and validation via a combination of peer review at a journal and additional evaluation by the wider scientific community.

**RONALD D VALE AND ANTHONY A HYMAN**

A scientist conducts experiments, analyzes the data from these experiments, and then arrives at a new understanding of the natural world. Once confident of their discovery, the scientist will then transmit this new knowledge to the rest of the scientific community. If the work is original and the evidence is accurate and compelling, it is expected that other scientists will give credit to the scientist who made the original discovery. However, this process of establishing "priority" for the discovery is not straightforward and disputes over priority have permeated every epoch of modern science.

In his classic article on the priority of discovery, Robert Merton describes how many of the early giants in physics and chemistry (Galileo, Newton, Hooke, Cavendish, Lavoisier, Watt and Faraday) were embroiled in battles over priority (*Merton, 1957*). Darwin's and Wallace's independent conception of natural selection as a driving force for evolution is a fascinating case study that reveals the complexities of priority, even when the scientists themselves act benevolently and respectfully. Debates over priority continue today, from the mildly aggravating "they should have cited my paper" to deliberations over jobs and prizes. Indeed, as long as human nature persists and the scientific enterprise places a premium on original work, then controversies and angst over priority will remain an inevitable component of science.

Here, we re-examine whether systems that have been used to establish priority in the past are well-suited to the present day. With many more scientists on the planet, all accessing the same information and many seeking to answer reasonably evident next questions, competitive situations arise more frequently. Furthermore, the speed of global communication has increased dramatically. These changes in science and communication technologies have led us to revisit two questions: how should a new discovery be communicated, and how should priority be established?

> **Establishing priority for a discovery is not straightforward and disputes over priority have permeated every epoch of modern science.**

## Patents and priority

Can scientists learn from the patent system with regard to defining priority? After all, the granting of a patent also involves assigning intellectual property based upon an original invention or discovery. A patent transfers knowledge into an asset that can be bought and sold; many scientists use patents to claim and protect their work for purposes of commercialization. However, the patent system and science work in very different ways. First, the granting of a patent is defined by a set of written guidelines. In contrast, scientific priority is guided by the culture and practices within a scientific community. Second, patent laws differ significantly in various countries, whereas the unwritten rules of scientific priority apply to all countries since knowledge is global. Third, the granting of a patent is binary; the patent is either granted or it is not, although this decision can be challenged in a court. In science, on the other hand, priority becomes evident through a process in which scientists credit one another's work in papers and at meetings; there are no appointed scientific inspectors or judges, with the exception of the very tiny fraction of scientific work being considered by prize committees. This means that it can take years for work to be evaluated and priority to be established.

There is another, even more fundamental, difference between the patent system and science. The primary goal of a patent is to stake territory as one's own and to *exclude others* (or make them pay for using your intellectual property), thereby serving the goal of commercialization. The invention only enters the public domain for free use after many years. In contrast, scientists publish their results to *encourage others* to make use of and build upon them. Indeed, priority can only be established when other scientists validate (i.e. replicate) the original discovery and affirm its importance (usually by building on it). We therefore need to look beyond the patent system for guidance.

## A two-step process for establishing priority

Merton made an interesting analogy between a discovery and personal property. When a scientist makes a discovery, at that moment, it belongs to the scientist and no one else. However, while this private eureka moment is a thrilling experience, exclusive ownership of the discovery is of little value, as it neither advances the scientist's reputation nor serves science as a whole. Thus, the scientist must eventually disclose the discovery and transfer the property to the scientific community at large. Once transferred, the discovery is no longer under the control of the original investigator, and all scientists have the right to make use of this new knowledge. Implicit in signing over his/her property, the scientist expects the scientific community to acknowledge their contribution. If the scientist does not receive this credit, then he/she feels that an injustice has occurred.

This relationship between the individual scientist and the scientific community highlights two steps in the establishment of priority. First, there is the transfer of knowledge from the scientist to the broad scientific community: we call this first step "disclosure". In a second step, the scientific community responds to the disclosure by assessing whether it is correct and of sufficient interest to merit attention and further development. It is this second step, which we call "validation", that establishes the scientist's reputation and results in rewards, such as career advancement and grants that enable more scientific work. However, unlike disclosure, which is a discrete time-stamped event, validation can take variable amounts of time, and the most novel discoveries often take the longest to be acknowledged by the community. Next, we discuss the role of these two steps in establishing priority in more detail.

## Step 1: Disclosure

A disclosure for establishing priority should involve a fair and complete transfer of knowledge from the scientist to the world-wide scientific enterprise. An acceptable method of disclosure will generally fulfill the following four criteria: (i) inclusion of all of the data along with a written interpretation of the data; (ii) a full description of the methodologies used, so that the work can be replicated and extended by other scientists; (iii) communication through a widely-recognized and stable venue (which must have mechanisms in place to ensure the permanence of the work); (iv) inclusion of a time stamp to indicate when the work was disclosed. What are the different options for disclosure?

### *Disclosure through a journal*

The publication of a paper in a peer-reviewed journal fulfills these four criteria. However, publishing in a journal also requires the scientist to hand over control of the timing of disclosure to the journal. Given the unpredictable nature of editorial rejection, peer-review, and the

publication process, the delay between the submission of a paper and its publication can range from a few weeks to more than two years. Furthermore, except for a handful of journals (such as BMJ Open, F1000Research and PeerJ), there is no public record of the original submission and the ways in which the manuscript changed during the peer-review process. The lack of such a record precludes using the date of the original submission as the date of disclosure. In addition, for papers published in subscription journals, disclosure is limited to those scientists who have access to those journals. Thus, while we later argue for the value of journals in validation, in their current form they slow down and create inequities in how knowledge is transferred from the scientist to the world-wide scientific community.

### Disclosure through a preprint server

A preprint server allows a scientist to post a completed study on the internet and immediately disclose the work without access barriers (*Vale, 2015*). A preprint does not undergo peer review, although many servers ensure that the study is scientific in nature. The preprint server arXiv, established in 1991 by Paul Ginsparg and now operated by Cornell University, is widely used in the physics, mathematics, and computational sciences communities. In recent years similar servers has been established for the life sciences, including bioRxiv (which is run by the Cold Spring Harbor Laboratory), PeerJ Preprints, and the quantitative biology section of arXiv.

A manuscript posted as a preprint could satisfy the four criteria listed above for disclosure, with certain qualifications. Since preprints are

**A preprint server allows a scientist to post a completed study on the internet and immediately disclose the work without access barriers.**

similar or identical in content to submitted journal manuscripts, criteria 1 and 2 can be met if the manuscript contains the necessary data, interpretation and details about methodologies. Whether criteria 3 and 4 are met depends upon the nature of preprint server, specifically

whether it has the ability maintain a permanent record and whether it is highly visible in the relevant scientific communities.

### Disclosure at a scientific meeting

Several decades ago, presenting work at a scientific meeting was often accepted by colleagues in a field as a reasonable mechanism for establishing priority. In these earlier days, however, the entire field of molecular biology could gather at a Cold Spring Harbor Meeting, which is no longer possible. Furthermore, the amount of data and methodologies presented in a meeting talk or poster is usually insufficient to meet criteria 1 and 2 and is generally not retained in the form of a permanent record (criteria 4).

While meetings fall short as a reliable mechanism for disclosure, we also recognize their substantial benefits. Meeting presentations allow scientists to get feedback on and subsequently improve their work, and are also outstanding training experiences for students and postdocs. Unfortunately, meetings are becoming increasingly filled with published or nearly published work. However, if preprints become widely accepted as a form of disclosure for the purposes of establishing priority, then more scientists might be more willing to share their work at meetings prior to journal publication.

### Speed versus quality

While the timing of disclosure is important for establishing priority, racing to be first at the expense of quality could spell disaster for a scientist, especially if he/she gets a reputation for rushing out low-quality work. Moreover, we believe that a strict, time-stamp-based "winner takes all" philosophy of priority (first articulated by Francois Arago, the secretary of the French Academy of Sciences, in the first half of the 19th century; *Strevens, 2003*; *Merton, 1957*) is not in the best interests of science. Moreover, it often falls short in practice as there are plenty of examples in which several groups have been co-credited for a particular discovery, even though their papers did not all appear at the same time. Furthermore, in the long run, the quality of the work is just as important as being first in the eyes of the scientific community.

Darwin and Wallace provide a case in point. Both scientists are recognized for their independent idea of natural selection and its role in evolving new species. But even in their lifetimes, it became broadly recognized, including by Wallace himself, that it was the outstanding corpus

of evidence in "On the Origin of Species" that associated Darwin's name most widely with the theory of evolution. Therefore, racing to disclose small and incomplete pieces of work is not a successful strategy for establishing priority, achieving recognition, or developing a good scientific reputation. Moreover, even in competitive situations, when time is of the essence, most scientists take into account the likely reaction of the scientific community to a paper.

## Step 2: Validation

If preprint servers can be used to disclose scientific work, what is purpose of a journal? The answer is that the disclosure of a discovery is of limited value to the scientist, unless the work is seen and discussed, analyzed for credibility and accuracy, repeated and cited by other scientists. Currently, journals and the peer-review process play a central role in the validation of scientific work. One reads a published journal article knowing that two or more colleagues have spent time carefully examining the work and looking for obvious errors in the experiments and/or interpretation. Widely-read journals also provide visibility for a scientific work, which is not achieved by the democracy of preprint servers. Indeed, an important function of an editor is to assemble and draw attention to an interesting set of papers in a particular field.

However, it is important to recognize that peer-review by a journal is a first step towards validation, rather than being the final word on it. Published papers in high-profile, peer-reviewed journals have been proven to be wrong or, in rare cases, to have been falsified; conversely, papers rejected after peer review have later

> **It is important to recognize that peer-review by a journal is a first step towards validation, rather than being the final word.**

achieved widespread recognition for changing the field. Therefore, establishing priority involves a broader scrutiny from the scientific community after publication. This process, as history has shown, can take years, and is often most visible in the form of citations. Important discoveries

that stand the test of time continue to be highly cited within a field, while those that were wrong or flawed generally fall by the wayside.

The traditional peer-review process used by journals is far from perfect. Furthermore, the heightened competition between scientists for space in high-profile journals (and the rewards that are thought to follow from publishing in these journals, such as grants and jobs) has strained the peer-review system, which seemed to work reasonably well until just a couple of decades ago (*Alberts et al., 2014*). Some have proposed that journals and peer review as currently practiced should be abandoned as soon as possible. We would argue against this: the need for a system of validation has only become more pronounced as the volume of scientific work has increased. Moreover, a number of journals are exploring ways to improve the peer-review process.

## Conclusions

In this article we have discussed how priority in science is established in two phases: disclosure and validation. In the life sciences, disclosure and the first stage of validation (peer review) are currently combined in the publication of a paper. However, science could be better served by first disclosing the work in a publically accessible forum and then, either simultaneously or at a later date, submitting the work to a journal for peer review. Disclosing manuscripts before they have been peer-reviewed would bring many benefits: the scientist would retain control over the timing of disclosure; other scientists would be able to provide feedback (positive or negative) before the work was published; other scientists would also be able to build on the work if they wished; and the public would benefit from the catalytic effects that ensue when knowledge is shared.

One way to implement earlier disclosure would be for journals themselves to post papers on their own website as soon as they are submitted, but this is unlikely to be universally implemented and may be awkward for authors if their paper is rejected. As discussed here and elsewhere, another option would be to use a preprint server. In February 2016, at the Accelerating Science and Publication in Biology (ASAPbio) meeting, the attendees – who included biologists (both junior and senior), funders and journals – concluded through discussion and private vote that preprints should constitute an acceptable means of disclosure for the purposes of establishing priority in the life

Vale and Hyman. eLife 2016;5:e16931. DOI: 10.7554/eLife.16931

sciences, and also be accepted as "evidence of achievement" when considering applications for grants, jobs and promotions (*ASAPbio, 2016*; *Berg et al., 2016*).

However, some biologists worry that submitting a preprint will lead to them being "scooped" if other scientists do not pay attention to work reported in preprints and do not cite them. What lessons might biologists learn from physicists regarding preprints and priority? arXiv has a very large intake of manuscripts (100,000 per year) and physicists have developed a habit of checking arXiv every morning to learn about the latest work in their field. Thus, the arXiv preprint server has become a highly visible venue for announcing new discoveries in physics. If preprint-based disclosure for priority is to take hold in the life and biomedical sciences, submissions will need to grow and preprints will need to be made more visible and more easily discoverable than they are now. In addition, trustworthy and community-led governance of preprints will be essential, and funding agencies and universities will need to put policies in place to accept preprints in applications for grants, jobs, and promotions.

Any preprint system must be coupled to a reliable system of validation. Journals have established an infrastructure for peer review, and they provide the present-day gold standard for validation. However, as noted above, peer review is far from perfect and must be combined with other forms of validation that take place over longer time scales and involve more than a handful of editors and referees. In the past few years, new forms of post-publication assessment, review and discussion have emerged, such as article-level metrics, blogs, social media and websites such as PubMed Commons and PubPeer (*Slavov, 2015*). In our opinion, a combination of preprints, journal-based peer review and new forms of post-publication validation would bring together the advantages of new digital technologies with what we value about traditional approaches.

**Ronald D Vale** is in the Department of Cellular and Molecular Pharmacology and the Howard Hughes Medical Institute, University of California, San Francisco, San Francisco, United States
Ron.Vale@ucsf.edu

**Anthony A Hyman** is in the Max Planck Institute of Molecular Cell Biology and Genetics, Dresden, Germany
hyman@mpi-cbg.de

***Competing interests:*** The authors declare that no competing interests exist.

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
