## [Decision Letter]

Thank you for submitting your article, "What defines 'priority of discovery' in the life sciences" to *eLife* for consideration as a Feature Article. Your article has been discussed by two Deputy Editors and Peter Rogers (the Features Editor).

I am writing to tell you that we will be willing to publish a shortened and revised version of the article.

Before publication, the revisions that need to be made are listed below:

1) The end of the article is somewhat underwhelming and needs to be beefed up. To do this please outline i) your recommendations for improving peer review; ii) your recommendations for improving how research is assessed after it has been peer-reviewed and published.

2) In the subsection "Disclosure through a journal", please be more explicit about the problems caused by the lack of a public record of the initial submission etc. Saying it is not possible to back-date etc is too vague.

3) Please say something about the ASAPBIO meeting, and maybe cite a report of that meeting.

4) In the Conclusions section please discuss the drawbacks of the proposal that journals post submissions before peer-review (ie, what happens to papers that are not sent to referees, and what happens to papers that are rejected are being refereed). Also, please mention that some journals (eg F1000 Research) already do something like this.

5) The original version of your essay was strongly in favor of the single preprint server model – is there a reason why this preference is not expressed in this version?

Finally, I propose deleting some sentences in the section on Validation (about competition and about elite/high-profile journals) that do not seem to belong where they are, but which you might want to include somewhere else in the article.

---

## [Author Response]

*The revisions that need to be made are listed below:*

1) The end of the article is somewhat underwhelming and needs to be beefed up. To do this please outline i) your recommendations for improving peer review; ii) your recommendations for improving how research is assessed after it has been peer-reviewed and published.

We have revised the final paragraph to incorporate an overall recommendation for post-publication peer-review, although kept it short in the interest of brevity.

2) In the subsection "Disclosure through a journal", please be more explicit about the problems caused by the lack of a public record of the initial submission etc. Saying it is not possible to back-date etc is too.

We have added a further explanation on this point.

*3) Please say something about the ASAPBIO meeting, and maybe cite a report of that meeting.*

We have now mentioned the ASAPbio meeting in the conclusion and reference a meeting report (to be published in a week).

*4) In the Conclusions section please discuss the drawbacks of the proposal that journals post submissions before peer-review (ie, what happens to papers that are not sent to referees, and what happens to papers that are rejected are being refereed). Also, please mention that some journals (eg F1000 Research) already do something like this.*

We have added a sentence of why we do not think journal posting upon submission will become wide-spread and also now reference F1000 Research as an example of such a platform.

*5) The original version of your essay was strongly in favor of the single preprint server model – is there a reason why this preference is not expressed in this version.*

We have added such a section back, but framed it more in terms of how a single preprint archive is need to enable the widespread visibility needed for priority of discovery.

Finally, I propose deleting some sentences in the section on Validation (about competition and about elite/high-profile journals) that do not seem to belong where they are, but which you might want to include somewhere else in the article.

We have re-added this sentence where it seems to fit (but still under Validation).

As final note, we kept the Abstract at essentially the same length. The Abstract as it stands provides a fairly complete summary of the main points of the article for those who will not read to read farther (which will be many). We also found similar Point of View article with similar or even longer abstracts.